# A Feasibility Study of Functional Lung Volume Preservation during Stereotactic Body Radiotherapy Guided by Gallium-^68^ Perfusion PET/CT

**DOI:** 10.3390/cancers15061726

**Published:** 2023-03-11

**Authors:** François Lucia, Mohamed Hamya, Fanny Pinot, Gaëlle Goasduff, Frédérique Blanc-Béguin, David Bourhis, Olivier Pradier, Anne-Sophie Lucia, Simon Hennebicq, Maëlle Mauguen, Romain Floch, Ulrike Schick, Vincent Bourbonne, Pierre-Yves Salaün, Pierre-Yves Le Roux

**Affiliations:** 1Radiation Oncology Department, University Hospital, 29200 Brest, France; 2LaTIM, INSERM, UMR 1101, University of Brest, 29200 Brest, France; 3Service de Médecine Nucléaire, CHRU de Brest, 29200 Brest, France; 4GETBO, INSERM, UMR1304, Université de Bretagne Occidentale, 29200 Brest, France

**Keywords:** lung cancer, stereotactic body radiation therapy, Gallium-68 perfusion PET/CT

## Abstract

**Simple Summary:**

Lung stereotactic body radiation therapy (SBRT) has become a real alternative treatment for inoperable stage I-II non-small cell lung cancer and lung oligometastases with a high local tumor control rate. However, SBRT remains associated with significant pulmonary toxicity. Perfusion positron emission tomography/computed tomography (PET/CT) with ^68^Ga-macroaggregated albumin (^68^Ga-perfusion) is a very attractive imaging tool for functional lung avoidance during radiotherapy planning. This single-center prospective study shows the feasibility of significantly decreasing the doses delivered to the lung functional volumes in the lung SBRT using ^68^Ga-perfusion PET/CT while still respecting target volume coverage and doses to other organs at risk.

**Abstract:**

The aim of this study was to assess the feasibility of sparing functional lung areas by integration of pulmonary functional mapping guided by ^68^Ga-perfusion PET/CT imaging in lung SBRT planification. Sixty patients that planned to receive SBRT for primary or secondary lung tumors were prospectively enrolled. Lung functional volumes were defined as the minimal volume containing 50% (FV50%), 70% (FV70%) and 90% (FV90%) of the total activity within the anatomical volume. All patients had a treatment planning carried out in 2 stages: an anatomical planning blinded to the PET results and then a functional planning respecting the standard constraints but also incorporating “lung functional volume” constraints. The mean lung dose (MLD) in functional volumes and the percentage of lung volumes receiving xGy (VxGy) within the lung functional volumes using both plans were calculated and compared. SBRT planning optimized to spare lung functional regions led to a significant reduction (*p* < 0.0001) of the MLD and V5 to V20 Gy in all functional volumes. Median relative difference of the MLD in the FV50%, FV70% and FV90% was −8.0% (−43.0 to 1.2%), −7.1% (−34.3 to 1.2%) and −5.7% (−22.3 to 4.4%), respectively. Median relative differences for VxGy ranged from −12.5% to −9.2% in the FV50%, −11.3% to −7.2% in the FV70% and −8.0% to −5.3% in the FV90%. This study shows the feasibility of significantly decreasing the doses delivered to the lung functional volumes using ^68^Ga-perfusion PET/CT while still respecting target volume coverage and doses to other organs at risk.

## 1. Introduction

Lung cancer is the leading cancer in terms of mortality, with more than 1.6 million deaths per year [1]. Non-small cell lung cancer (NSCLC) accounts for 80% to 85% of all lung cancers [1]. Although ablative surgery is the gold standard treatment for patients with operable early-stage NSCLC, lung stereotactic body radiation therapy (SBRT), particularly for patients who are medically inoperable, has become a real alternative treatment [2].

For patients with distant metastases secondary to solid tumors, systemic treatment remains the therapeutic standard [3]. However, in the event of oligometastatic disease or oligorecurrence, lung SBRT is often used on the metastatic sites [4,5,6,7,8,9]. 

In these indications, local tumor control rate is higher than 90%. However, lung SBRT remains associated with significant pulmonary toxicity. Grade ≥ 3 pulmonary complications are reported in 10% of cases and grade ≥ 2 in up to 20% of cases [2]. Unfortunately, there is no conventional dosimetric parameter that can predict pulmonary toxicity following lung SBRT [10]. Therefore, a current major preoccupation is to preserve lung function and to reduce radiation induced lung injury (RILI) [10].

During standard lung SBRT planning, dose constraints are defined by the anatomical lung volume [10,11]. This planning considers the lung functionally uniform and does not take into account the variability of regional lung function distribution, especially in patients with tobacco related lung diseases and cancer treatment induced pulmonary toxicities. Mapping the distribution of regional pulmonary function could be useful for optimizing SBRT treatment plans by limiting the dose received by functional territories to the detriment of poorly or non-functional areas in the hope of reducing pulmonary toxicity [12,13].

Previous studies using lung perfusion scintigraphy, especially with single photon emission computed tomography (SPECT) technique, have shown a radiation dose-effect relationship in functional regions of the lung in patients undergoing radiation treatment for lung, breast cancers and lymphoma [12]. However, SPECT has inherent technical limitations, mostly relative to poor spatial and temporal resolution, preventing an accurate and reproducible mapping of lung functional volumes [14,15]. As a consequence, functional lung avoidance planning using SPECT imaging has not yet been adopted in daily clinical practice.

Lung perfusion PET/CT imaging is a novel imaging technique for regional lung function evaluation with greater sensitivity and better spatial and temporal resolutions [16,17]. Similar to conventional lung perfusion scintigraphy, images are obtained after intravenous administration of macro-aggregated albumin (MAA) particles radiolabeled with Gallium-68, a ß+ isotope, instead of Technetium-99m as scintigraphy, which embolize in the pulmonary capillaries according to pulmonary blood flow [18]. This offers the opportunity to improve the accuracy of lung functional volumes delineation and its integration in thoracic radiation therapy planning. Siva et al. demonstrated the feasibility of preserving lung functional volumes in a cohort of 14 patients treated with 3D conformal radiotherapy for NSCLC adapted to ^68^Ga-perfusion PET/CT [19]. However, SBRT is a more precise and conformal technique that allows the irradiation of smaller volumes of organs at risk (OARs) [2]. To date, no study has evaluated the performance of ^68^Ga-perfusion PET/CT imaging to personalize lung SBRT planning and dosimetry.

The aim of this study is to assess the feasibility of sparing functional lung areas by integration of pulmonary functional mapping guided by ^68^Ga-perfusion PET/CT in lung SBRT planification.

## 2. Materials and Methods

### 2.1. Study Design and Participants

The PEGASUS trial is a single-center prospective study. The eligible study population consisted of patients aged > 18 years that planned to be treated in the radiotherapy department at the Brest University Hospital, France, with SBRT for primary or secondary lung tumors. Exclusion criteria included the inability to give informed consent, patients under guardianship or curatorship, pregnant or breastfeeding women and contra-indication to the administration of human albumin macroaggregates.

The study was approved by the Nord Ouest IV Ethics Committee (ID RCB: 2021-002224-20) and registered in ClinicalTrial.gov registry (NCT04942275). Written informed consent was obtained from all participants.

### 2.2. PET/CT Protocol

All patients underwent lung perfusion PET/CT scan acquired on a digital Biograph Vision 600 PET/CT scanner (Siemens Healthineers, Knoxville, TN, USA). [^68^Ga]Ga-MAA suspensions were prepared in the radiopharmacy using an automated process with a miniAIO^®^ module and disposable cassettes from Trasis (Liège, Belgium). The fully automated process was performed from a commercial MAA kit used for 99mTc labelling (the commercial [^99m^Tc]Tc-MAA kits (Pulmocis^®^) were purchased from CIS Bio International (CURIUM, Saclay, France)). Details on [^68^Ga]Ga-MAA preparation and protocol acquisition are provided in Appendix B.

The lung volumes were contoured using MIMimage analysis software (MIM 7.2.3; MIMSoftware, Cleveland, OH, USA). An automatic contouring of the whole lung anatomical volume (AV) based on Hounsfield unit value was initially performed and then visually adjusted to match normal contours if required. Within the AV, lung functional volumes (FV) were defined using an adaptative-absolute-threshold algorithm delineating the minimal volume containing 50% (FV50%), 70% (FV70%) and 90% (FV90%) of the total activity within the AV, respectively. Figure 1 shows an example of lung functional volume delineation. These volumes then were imported as DICOM structures into a Pinnacle V16.2 planning system.

### 2.3. Radiation Therapy

All patients were planned to receive three fractions of 18 Gy each (total 54 Gy), or if the tumour was less than 2 cm from the chest wall, four fractions of 12 Gy each (total 48 Gy) or if the tumor was central or ultracentral, eight fractions of 7.5 Gy each (total 60 Gy). Radiotherapy planning scans were performed with patients positioned supine with both arms elevated above the head. A respiratory-sorted 4-dimensional computed tomography (4DCT) data set was generated using the planning CT (Siemens, Somatom, Munich Germany) coupled with the Varian real-time position management (RPM) gating system. The 4DCT scans were acquired in helical mode and binned into 8 phases for image reconstruction. From the respiratory-sorted imaging phases, average (AVG) and maximum intensity projection (MIP) series were reconstructed. The internal target volume was delineated on the basis of a four-dimensional CT planning scan to take account of tumour motion on a MIM workstation. The internal target volume was expanded 3 mm in all plans to create the planning target volume. The lung organ at risk (OAR) was defined as the volume of both lungs minus the volume of the internal target volume (ITV). Treatment planning was performed on the Pinnacle V16.2 planning system using the AVG series. Treatment was prescribed such that 99% of the planning target volume (PTV) received at least 99% of the prescribed dose. The maximum dose at the PTV should be <140% of the prescribed dose. Dose constraints to OARs respected the international guidelines [20]. All treatments used 6-MV photons. Dose distributions were performed with volumetric modulated arc therapy (VMAT). Pulmonary heterogeneity has been taken into account by using the Collapse Cone Convolution algorithm. Treatment quality was verified by calculating the conformality index (CI) (a ratio of prescription isodose volume to the PTV), a ratio of 50% isodose volume to the PTV (R50%) and the maximum dose at 2 cm from the PTV in any direction as % of prescribed dose (D2cm) (Appendix A). Lung SBRT was performed using a Novalis Tx^®^ linear accelerator equipped with Varian RPM gating system (Varian Medical Systems, Palo Alto, CA, USA). Image guidance with cone beam CT was used to confirm the position of the tumor immediately before each treatment.

All patients had a treatment planning carried out in 2 stages. First, an anatomical planning was carried out, blinded to the PET results. Then, a functional planning, respecting the standard constraints applied during anatomical planning, but also incorporating “lung functional volume” constraints defined by pulmonary PET (FV50%, FV70% and FV90%), was carried out.

### 2.4. Data and Statistical Analysis

Dose to OAR and target volumes were computed for both treatment plans. The mean lung dose (MLD) in functional volumes (FV50%, FV70% and FV90%) using the anatomical and functional plans were calculated and compared using the paired samples Wilcoxon test. The percentage of lung volumes receiving 5, 10, 15 and 20 Gy (V5, V10, V15 and V20Gy) within the lung functional volumes (FV50%, FV70% and FV90%) were also calculated and compared using the paired samples Wilcoxon test. Median absolute (dosimetric parameter of functional plan minus dosimetric parameter of anatomical plan) and relative (dosimetric parameter of functional plan minus dosimetric parameter of anatomical plan divided by dosimetric parameter of anatomical plan) differences and their respective range were reported. Results were also analyzed in patients with and without preexisting airway disease (FEV1/FVC < 0.7 or DLCO < 60%) [21]. All analyses were done using the R++ platform for statistical programming, version 1.5.03.

## 3. Results

### 3.1. Study Population

Between July 2021 and January 2022, 60 consecutive patients with primary or secondary lung tumors were enrolled into this prospective study. The median (range) age of patients was 69 (51–84) years old. Patient characteristics and dose target volumes are outlined in Table 1. Forty-three (72%) patients consisted of current or previous smokers and 19 (32%) patients had previously undergone thoracic radiotherapy. Forty-eight patients had full respiratory function tests. Of these 48 patients, 31 (65%) had a preexisting airway disease (FEV1/FVC < 0.7 or DLCO < 60%) [21]. The median (range) ITV volume and PTV volume were 6.4 cc (1.0–83.4 cc) and 14.4 cc (3.2–139.7 cc), respectively.

### 3.2. Dosimetric Comparisons

For all patients, both anatomical and functional plans met the OARs and target dose constraints (Appendix A). There was a statistically significant difference between the 2 plans regarding PTV coverage, 99.6% (71.8–100%) vs. 99.6% (70.2–100%) (*p* = 0.01), but with similar ITV coverage 100% (83.4–100%) vs. 100% (83.3–100%) (*p* = 0.38). Only one patient had insufficient PTV and ITV coverage for both treatment plans to meet the constraints at the main bronchi in the case of an ultra-central tumor. Mean relative volume of FV50%, FV70% and FV90% was 32% (SD 3%), 50% (SD 4%) and 75% (SD 4%) of the AV.

MLD in functional volumes are presented in Table 2. SBRT lung planning optimized to spare lung functional regions led to a significant reduction (*p* < 0.0001) of the MLD in functional volumes.

Median absolute difference of the MLD in the FV50%, FV70%, FV90% and anatomical volumes (AV) between the anatomical and functional plans was −0.2 Gy for FV and −0.1 Gy for AV (See Table 2). Twelve (20%) patients had a decrease of at least 0.5 Gy of the mean dose in the FV50% (See Figure 2). 

Median relative difference of the MLD in the FV50%, FV70%, FV90% and AV was −8.0% (−43.0 to 1.2%), −7.1% (−34.3 to 1.2%), −5.7% (−22.3 to 4.4%), and −4.7% (−15.8 to 8.5%), respectively (See Table 2). Twenty-six (43%) patients had a decrease of at least 10% of the MLD in the FV50%, and six (10%) patients a decrease of at least 20% (See Figure 2). 

For 17 patients without a preexisting airway disease, median relative difference of the MLD in the FV50%, FV70% and FV90% was −8.2% (−20.7 to 1.2%), −7.6% (−18.6 to 1.2%) and −5.7% (−16.2 to 4.4%). For 31 patients with a preexisting airway disease, median relative difference of the MLD in the FV50%, FV70% and FV90% was −7.5% (−43.0 to 0%), −6.9% (−34.3 to 0%) and −5.2% (−22.3 to 1.3%). There was no significant difference between the 2 subgroups (*p* = 0.23, *p* = 0.32 and *p* = 0.59, respectively).

SBRT lung planning optimized to spare lung functional areas led to a significant reduction (*p* < 0.0001) of the V5 to V20 Gy in all functional volumes (FV50%, FV70% and FV90%) (See Table 3 and Appendix A). Median relative differences ranged from −12.5% to −9.2% in the FV50%, −11.3% to −7.2% in the FV70%, −8.0% to −5.3% in the FV90%, and −6.4% to −2.9% in the anatomical volume. Most important reduction was observed in highly functional lung volumes (FV50%) and for the V5 and V10Gy. Thirty-four (57%) patients had a decrease of at least 10% of the V5Gy within the FV50% volume, and 16 (27%) a decrease of at least 20%. 

Figure 3 demonstrates a representative example of personalized SBRT lung planning and dosimetry guided by ^68^Ga-perfusion PET/CT imaging.

## 4. Discussion

This is the first study incorporating the use of perfusion-PET in lung SBRT planning and demonstrating proof of principle that this technology may be used for functional lung avoidance. Our study demonstrated a statistically significant improvement (*p* < 0.0001) in dose to functional lung volumes using functionally adapted radiotherapy plans while still respecting target coverage or dose constraints to other OARs. Indeed, the MLD to the functional volumes was reduced by 6 to 8% and the irradiated volumes by 5 to 12%.

As expected, the absolute gains obtained with SBRT, around 0.2 Gy, were lower than with conformal techniques due to lower healthy lung volume irradiated in SBRT [2]. In a recent meta-analysis that included 19 studies that used a wide range of functional imaging techniques and functional lung volume definitions, the mean functional lung dose was reduced by 1.98 Gy (95% CI 0.57–3.39) [12]. None of them used SBRT. Siva et al. were the first and only so far that assessed lung PET/CT imaging for functional lung avoidance. In a series of 14 patients with lung cancer irradiated in conformal technique, they demonstrated that PET-guided functional planning allowed a reduction by a median of 0.86 Gy of the MLD in well perfused lungs [19]. However, the gains obtained on some functional lung volumes were at the expense of other volumes, for example, the significant decrease in functional V60 resulted in a significant increase in functional V5Gy. In our study, the optimization allowed a significant decrease in the overall functional lung volumes irradiated, while respecting the target volume coverage and the OARs dose constraints. 

On the other hand, the relative gains obtained with SBRT were in line with studies using conformal techniques. For instance, the mean functional volume receiving 20 Gy was reduced by 4.42% (95% CI 1.66% to 7.18%) in the meta-analysis of 19 studies mentioned above, while it ranged from 5.3% to 9.2% in our series [12].

Furthermore, if these overall results demonstrate the global performance of the technique, it is also important to note the major reduction of the dose in lung functional volumes in some individual patients. Thus, 10% of patients had a decrease greater than 20% of the MLD in the FV50%, and 27% had a decrease greater than 20% of the V5Gy within the FV50% volume. In that respect, we performed the analysis in patients with and without preexisting airway disease (FEV1/FVC < 0.7 or DLCO < 60%), on the assumption that patients with pulmonary disease are more likely to have heterogenous lung function distribution and thus to benefit from functionally adapted radiotherapy. However, we did not observe a major difference between both groups and found that optimization of radiation treatment plans was also possible in patients without preexisting pulmonary condition.

Lung functional sparing is an increasing concern because SBRT has become the gold standard RT technique for stage I-II NSCLC and for secondary lung tumors in oligometastatic patients [2,4,6,7,8,9]. Although SBRT leads to less pulmonary toxicity than conventional RT, the rate of grade 2+ RILI remains significant, between 9 to 30% [2,10]. Patients with interstitial lung disease were at particular risk of toxicity [10], highlighting the need to find pulmonary function-sparing methods. In addition, SBRT is indicated as an alternative to surgery in patients whose lung function is already impaired with heterogeneous lung function distribution and greater sensitivity to even minimal loss of lung capacity [2]. Finally, the development of cancer treatments (targeted therapies, immunotherapy) [3] has led to an improvement in patient survival but also to an increase in the number of recurrences and indications for re-irradiation, particularly with a highly conformal technique such as SBRT [22]. Thus, a better sparing of the pulmonary functional areas with each course of SBRT could lead to higher possibilities of re-irradiation or to limit the risks of toxicities of these re-irradiations [22].

Perfusion PET/CT imaging is a very attractive test for functional lung avoidance during radiotherapy planning [13,14,17]. Besides increasing the accuracy of lung functional volumes delineation owing to the use of PET technology, the test offers many advantages. Perfusion PET imaging is a simple and non-invasive imaging modality, with no contraindication or side-effects. No special procedure such as fasting or diet is required. Images are acquired immediately after administration of the radiotracer, and the acquisition time is approximately 5 min. Furthermore, one synthesis production of ^68^Ga-MAA allows one to perform ^68^Ga-perfusion PET/CT for 6 patients, making the test very convenient to implement in a daily PET/CT program. This advantage is especially evident during the hour (uptake time) between the first injections of ^18^F-FDG, the most-used tracer in oncology, and its first acquisition on the PET tomograph. Finally, ^68^Ga-MAA is very convenient radiotracer for clinical use, and ^68^Ga generators are increasingly available in nuclear medicine departments due to growing use of ^68^Ga-based tracers for prostate cancer and neuroendocrine tumour imaging.

Our study has several limitations. Firstly, we only assessed the feasibility of personalizing radiation planning to lung functional mapping using ^68^Ga-perfusion PET/CT. However, it would be valuable to evaluate the prognostic impact in terms of RILI of doses delivered to the lung functional volumes. Indeed, a meta-analysis including 6 studies showed that the dose to functional volume had better predictive values than the dose to anatomical volume, including V20, V5 and MLD [12], but all these studies used SPECT, and none of the studies was statistically significant. Secondly, by adding “lung functional volume” constraints, the functional planning led to a significant reduction of the dose to the functional volumes but also to the lung anatomical volume. Nevertheless, there was a greater improvement in the dose to the functional volumes as compared with that of the anatomical lung, showing that the optimization is not blinded but functionally adapted. Of note is that the reduction of the dose in the AV was not reported in most of studies assessing imaging techniques for lung functional avoidance [12]. Thirdly, we aimed to deliver the same doses to the OARs for the functional planning as for the anatomical planning. However, doses to OAR were significantly lower than the constraints in the most cases, leaving the possibility of improved optimization to further preserve lung functional volumes. Finally, the improvements obtained in the lung functional volume resulted in a significant decrease in the PTV coverage (99.6 vs. 99.6, *p* = 0.01). However, the PTV coverage was respected with similar ITV coverage between the 2 treatment plans.

## 5. Conclusions

This is the first study to evaluate and demonstrate the benefits of ^68^Ga-perfusion PET/CT in the lung SBRT planning process in preserving functional lung volumes while respecting target volume coverage and doses to other OARs.

## Figures and Tables

**Figure 1 cancers-15-01726-f001:**
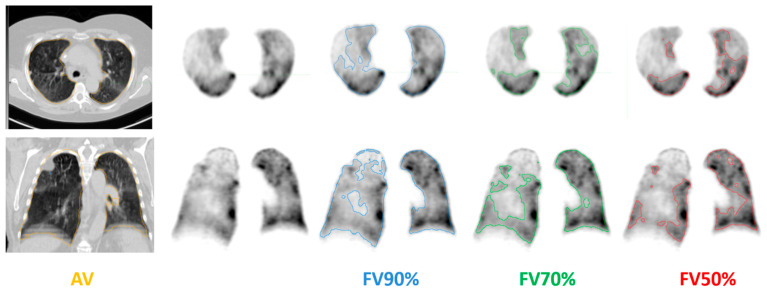
Example of anatomic volume (AV) (at the left) and lung functional volumes (FV) delineation. The FV90% (in blue), FV70% (in green) and FV50% (in red) volumes were defined as the minimal volume containing 90% (FV90%), 70% (FV70%) and 50% (FV50%) of the total activity within the AV.

**Figure 2 cancers-15-01726-f002:**
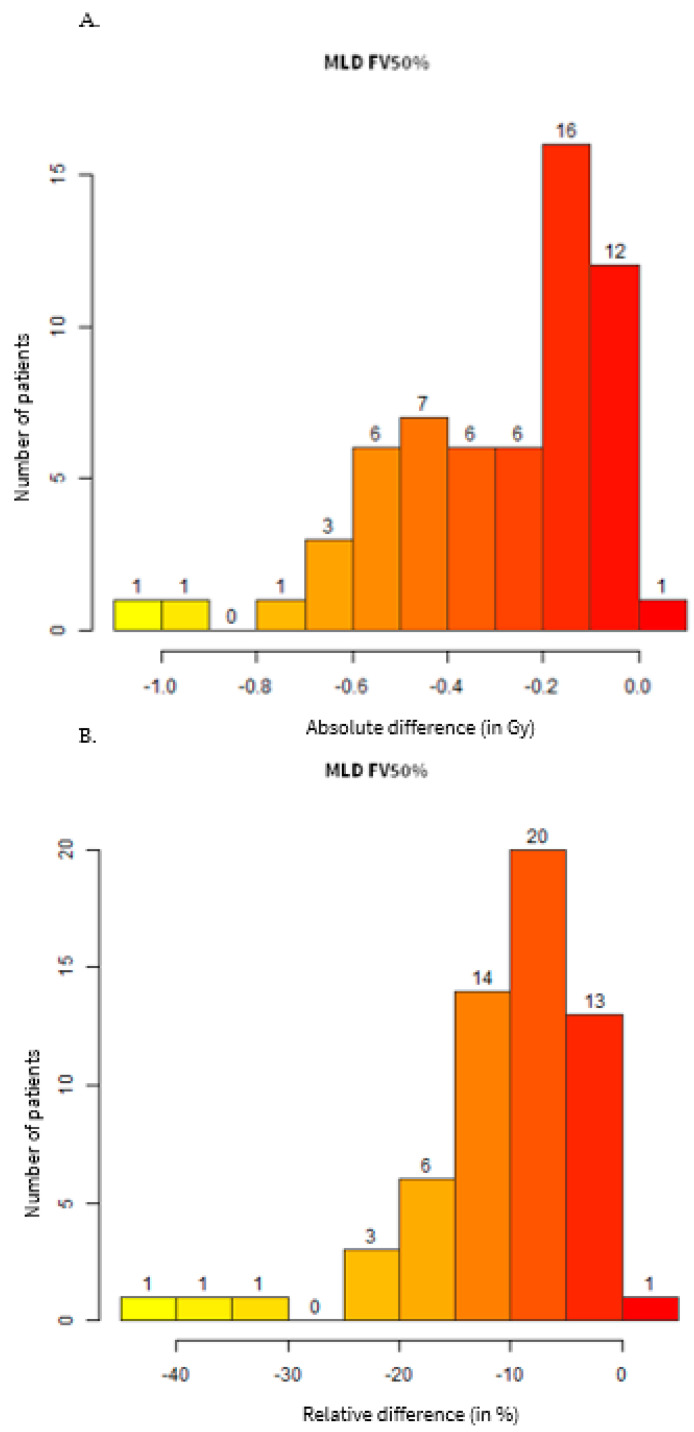
Distribution of absolute difference (**A**) and relative difference (**B**) of the mean lung dose (MLD) and relative difference for the V5Gy (**C**) between anatomical planning and functional planning in the FV50% lung functional volume.

**Figure 3 cancers-15-01726-f003:**
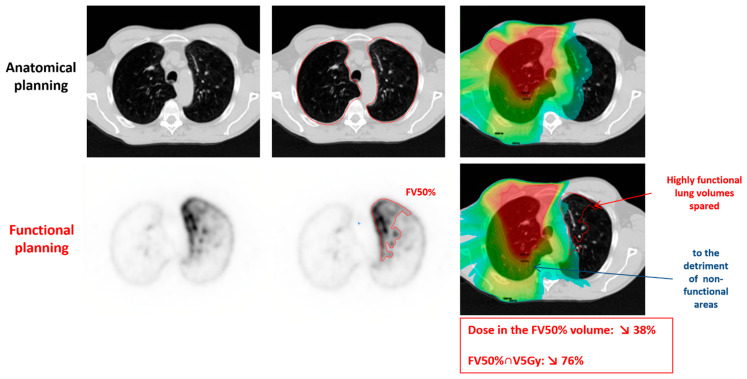
Example of personalized lung SBRT planning and dosimetry based on 68Ga-perfusion PET/CT imaging. The first row shows the conventional anatomical planning based on CT imaging. The second row shows functional planning based on 68Ga-perfusion PET/CT imaging. The anatomical planning included highly functional lung volumes in the left upper lobe. By incorporating an additional “functional lung volume constraint” to the lung FV50% in addition to the standard constraints in SBRT planning, it was possible to spare this highly functional volume in the left upper lobe, to the detriment of the posterior aspect of the right upper lobe which was non-functional. This resulted in a 38% decrease of the dose in the FV50% volume and a 76% decrease of the FV50% volume receiving more than 5Gy.

**Table 1 cancers-15-01726-t001:** Patients’ characteristics.

	Group	
	N = 60	%
Gender		
Male Female	31 29	52 48
Age median (range)	69 (51–84)	
Histology		
NSCLC Secondary lung tumor Without	25 23 12	42 38 20
Current smokers		
Yes No	19 41	32 68
Current or previous smoker		
Yes No	43 17	72 28
Prior thoracic RT		
Yes No	19 41	32 68
Number of lesions		
Single Multiple (2–3)	51 9	85 15
Prior thoracic surgery		
Yes No	14 46	23 77
PS		
0 1 2	27 25 8	45 42 13
FEV1 median (range)	66 (11–94)	
Missing	10	17
FEV1/FVC median (range)	67 (24–100)	
Missing	10	
DLCO median (range)	56 (9–119)	
Missing	19	32
ITV volume median (range)	6.4 cc (1.0–83.4)	
PTV volume median (range)	14.4 cc (3.2–139.7)	

Abbreviations: NSCLC = Non-Small Cell Lung Cancer; RT = Radiation Therapy; PS = Performance Status; FEV1 = Forced expiratory volume in one second; DLCO = diffusing capacity of the lung for carbon monoxide; ITV = Internal Target Volume; PTV = Planning Target Volume.

**Table 2 cancers-15-01726-t002:** Comparison of the mean lung dose to functional and anatomical volumes between the Anatomical Plan and the Functional Plan.

	Anatomical Plan	Functional Plan	Functional Plan Minus Anatomical Plan	Functional Plan Minus Anatomical Plan	Difference (*p*-Value)
			Absolute Difference	Relative Difference	
	Median (Range)	Median (Range)	Median Gy (Range)	Median % (Range)	
FV50%	3.1 (0.2–12.7)	3.0 (0.2–12.4)	−0.2 (−1.1 to 0.1)	−8.0 (−43.0 to 1.2)	<0.0001
FV70%	3.3 (0.3–10.7)	3.0 (0.3–10.4)	−0.2 (−0.8 to 0.1)	−7.1 (−34.3 to 1.2)	<0.0001
FV90%	3.0 (0.6–9.9)	2.8 (0.6–9.5)	−0.2 (−0.8 to 0.2)	−5.7 (−22.3 to 4.4)	<0.0001
Anatomical	2.7 (0.8–7.5)	2.5 (0.7–7.2)	−0.1 (−0.5 to 0.3)	−4.7 (−15.8 to 8.5)	<0.0001

**Table 3 cancers-15-01726-t003:** Comparison of the percentage of lung volumes receiving 5, 10, 15 and 20 Gy (V5, V10, V15 and V20) within the lung functional volumes (FV50%, FV70% and FV90%) and anatomical volume (AV) between the Anatomical Plan and the Functional Plan.

		Anatomical Plan	Functional Plan	Absolute Difference	Relative Difference	Difference (*p*-Value)
		Median (Range)	Median (Range)	Median (Range)	Median % (Range)	
FV50%	V5Gy	16.1% (0.1% to 48.2%)	14.0% (0% to 46.8%)	−1.4% (−8.4% to 4.5%)	−11.4% (−100% to 19.2%)	<0.0001
	V10Gy	8.6% (0% to 41.2%)	7.7% (0% to 40.0%)	−0.8% (−5.1%to 0.9%)	−12.5% (−99.9% to 6.5%)	<0.0001
	V15Gy	5.5% (0.0% to 30.5%)	4.7% (0.0% to 29.5%)	−0.5% (−5.3% to 0.3%)	−10.6% (−90.4% to 7.5%)	<0.0001
	V20Gy	3.6% (0.0% to 22.6%)	3.3% (0.0% to 21.8%)	−0.3% (−2.9% to 0.3%)	−9.2% (−100% to 9.2%)	<0.0001
FV70%	V5Gy	15.2% (0.3% to 9.9%)	13.5% (0.1% to 47.9%)	−1.3% (−6.6% to 3.6%)	−10.4% (−76.5% to 16.1%)	<0.0001
	V10Gy	8.7%% (0% to 34.9%)	7.5% (0% to 32.1%)	−0.7% (−4.1%to 0.5%)	−11.3% (−99.9% to 12.7%)	<0.0001
	V15Gy	5.7% (0.0% to 22.8%)	4.9% (0.0% to 22.0%)	−0.4% (−4.4%to 0.1%)	−10.0% (−93.3% to 13.9%)	<0.0001
	V20Gy	3.6% (0.0% to 16.7%)	3.3% (0.0% to 16.2%)	−0.2% (−2.1% to 0.3%)	−7.2% (−100% to 22.2%)	<0.0001
FV90%	V5Gy	14.0% (1.2% to 49.2%)	13.0% (1.0% to 46.7%)	−1.0 (−5.3% to 3.4%)	−7.9% (−42.8% to 16.6%)	<0.0001
	V10Gy	8.5% (0.5% to 32.0%)	7.5% (0.3% to 29.7%)	−0.6% (−3.7%to 0.5%)	−8.0% (−37.8%to 5.9%)	<0.0001
	V15Gy	5.7% (0.2% to 20.9%)	4.9% (0.2% to 19.8%)	−0.4% (−3.2%to 0.3%)	−7.7% (−41.8% to 11.5%)	<0.0001
	V20Gy	3.5% (0.2% to 15.0%)	3.2% (0.1% to 14.4%)	−0.1% (−1.5% to 0.2%)	−5.3% (−38.8% to 13.3%)	<0.0001
AV	V5Gy	13.6% (3.5% to 43.0%)	12.2% (2.8% to 40.7%)	−0.7 (−4.7% to 3.4%)	−6.0% (−23.9% to 18.0%)	<0.0001
	V10Gy	7.6% (1.9% to 27.1%)	7.0% (1.9% to 25.5%)	−0.5% (−3.0% to 0.7%)	−6.4% (−31.0% to 6.6%)	<0.0001
	V15Gy	4.7% (1% to 17.5%)	4.3% (1.0% to 16.8%)	−0.3% (−2.2% to 0.4%)	−5.1% (−41.2% to 11.2%)	<0.0001
	V20Gy	3.1% (0.7% to 12.1%)	3.0% (0.6% to 11.7%)	−0.1% (−1.3% to 0.5%)	−2.7% (−25.5% to 13.3%)	<0.0001

## Data Availability

Research data are stored in an institutional repository and will be shared upon request to the corresponding author.

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
