# Peer review of "A Feasibility Study of Functional Lung Volume Preservation during Stereotactic Body Radiotherapy Guided by Gallium-68 Perfusion PET/CT"

_cancers, 2023, doi:10.3390/cancers15061726_

Round 1

Reviewer 1 Report (Previous Reviewer 2)

Thank you for addressing the statistical issues I raised in my initial review of your manuscript submission.

Author Response

We thank the reviewer for this positive assessment of our work.

Reviewer 2 Report (Previous Reviewer 1)

Just a few comments in the attached file

Congrats for your excellent work!

Author Response

We thank the reviewer for this positive assessment of our work.

Comments

Please write the radiopharmaceutical always in the same way 68Ga-macroaggregated albumin or [68Ga]Ga-MacroAggregated Albumin

 Line 18: ….. 68Gamacroaggregated albumin ….. write 68Ga-macroaggregated albumin

We have made the correction

Line 24: …..guided by Gallium-68 (68Ga)-perfusion PET/CT imaging write the isotope only once as Gallium-68 or 68Ga

We have made the correction

Line 78: …..and not Technetium-99m as scintigraphy, …... please write: “instead of Technetium-99m used for scintigraphy,”

We have made the correction

Line 104: “[68Ga]Ga-MacroAggregatted Albumin (MAA)” write: [68Ga]Ga-MAA or [68Ga]Ga-MacroAggregated Albumin (without the acronym that has been already cited at line 77)

We have made the correction

line 106-107: ….. from a commercial MAA kit used for 99mTc labelling. (insert the kit brand i.e Pulmocis, Curium or similar)

We have added the kit brand

The commercial [99mTc]Tc-MAA kits (Pulmocis®) were purchased from CIS Bio International (CURIUM, Saclay, France)

Appendix A

Line 360: [68Ga]Ga-MacroAggregatted Albumin (MAA) write: [68Ga]Ga-MAA or [68Ga]Ga-Macroaggregated Albumin (only one “t”)

We have made the correction

Line 365: Controls performed …. write: Results of quality controls performed.....

We have made the correction

line 366: ...particles < 3μm : <2.2%

We have removed the colon

line: 362: …..from a commercial MAA kit used for 99mTc labelling …..(insert the kit brand i.e Curium or similar)

We have added the kit brand

 Line 371: Approximately 50 MBq of 68Ga-MAA were intravenously administrated this sentence follows the CT acquistion method: please move it soon after tracer preparation (after line 367)

line 373, Perfusion images were acquired immediately after. What do you mean? After tracer injection or after CT)? Please specify

We have added the explanation

« Perfusion images were acquired immediately after CT »

line 373: Perfusion PET data was …. write “were” delete “was”

We have made the correction

line 375: PET data was reconstructed … write “were” delete “was”

We have made the correction

This manuscript is a resubmission of an earlier submission. The following is a list of the peer review reports and author responses from that submission.

Round 1

Reviewer 1 Report

I would like to congratulate with Authors  for the innovative use of PET technique in SBRT planning  for early-stage NSCLC and particularly for the assessment of lung functional volumes for adapted radiotherapy.

Their road map in developing and validating both tracer production (68Ga-MAA) and image acquisition it's of particular value as a working method.

There are some comments in the attached file.

Author Response

Comments

Please write the isotopes always in the same way: Gallium-68 and Technetiun-99m (in extended) or 68Ga and 99mTc (symbol) : We have written the same way everywhere  “68Ga-perfusion PET/CT”

• i.e. Line 39, 37, 19, 21, 86, 89, 77 Please, choose how do you prefer to name the perfusion PET technique and use always the same

• Line 23: Gallium-68 perfusion PET/CT

• Line 19: Ga68 perfusion PET/CT

• Line 37, 86,89, 214, 304, 329: 68Ga perfusion PET/CT

• Line 39: Gallium68 perfusion PET/CT

• Line 236: 68Ga-MAA PET/CT imaging Perhaps you can call the technique 68Ga-perfusion PET/CT

Please define exactly the used radiopharmaceutical: it appears only in line 234 as 68Ga-MAA (fig. 3), it is never reported before in the text Perhaps it may be inserted in the Appendix A. It would be interesting if you have any data about quality controls of the prepared tracer.

As proposed, this was added in the Appendix A as follows :

“[68Ga]Ga-MacroAggregatted Albumin (MAA) suspensions were prepared in the radiopharmacy using an automated process with a miniAIO® module and disposable cassettes from Trasis (Belgium). The fully automated process was performed from a commercial MAA kit used for 99mTc labelling suspended in sodium acetate solution and a 68Ga eluate without pre purification as previously described by Blanc-Béguin et al.  The obtained suspensions were GMP compliant. Controls performed on [68Ga]Ga-MAA suspensions from independent syntheses were : radiochemical purity > 98.6%, particles < 3µm : <2.2%, radionuclidic purity > 99,999%, pH = 4, endotoxins level < 5 IU/mL and all suspensions were sterile.”

Line 17-20: clarify in the first statement which type of perfusion imaging allows functional lung avoidance

We have clarified the first statement.

“Perfusion positron emission tomography/computed tomography (PET/CT) with 68Gamacroaggregated albumin (68Ga-perfusion) is a very attractive imaging tool for functional lung avoidance during radiotherapy planning. This single-center prospective study shows the feasibility of significantly decreasing the doses delivered to the lung functional volumes in the lung SBRT using 68Ga-perfusion PET/CT while still respecting target volume coverage and doses to other organs at risk”

Line 14: primary OR secondary (delete “of”)

We have corrected this mistake

Line 14: write 60 in letters

We have corrected this mistake

Line 25-29: reverse the two sentences

We have reversed the 2 sentences

• All patients had a treatment planning carried out in 2 stages: an anatomical planning blinded to the PET results, then a functional planning, respecting the standard constraints, but also incorporating "lung functional volume" constraints. Lung functional volumes were defined as the minimal volume containing 50% (FV50%), 70% (FV70%) and 90% (FV90%) of the total activity within the anatomical volume.

Line 73-80: redundant sentences , can you simplify?

We have simplified this part

« Lung perfusion PET/CT imaging is a novel imaging technique for regional lung function evaluation with greater sensitivity and better spatial and temporal resolutions [16, 17]. Similar to conventional lung perfusion scintigraphy, images are obtained after intravenous administration of macro-aggregated albumin (MAA) particles radiolabeled with Gallium-68, a ß+ isotope and not Technetium-99m as scintigraphy, which embolize in the pulmonary capillaries according to pulmonary blood flow [18]. »

Line 77-78: please write isotopes in the same way (Technetium-99m, Gallium-68)

We have written isotopes in the same way

line 82: “Siva et al demonstrated the feasibility of preserving lung functional volumes in a cohort of 14 patients treated with 3D conformal radiotherapy for NSCLC. ….. • how they preserved? With SPECT or PET? You can change the sentence i.e. “Siva et al demonstrated the feasibility of preserving lung functional volumes in a cohort of 14 patients treated with 3D conformal radiotherapy for NSCLC adapted to 68Ga-perfusion PET/CT” or “adapted to PET-guided functional mpping” …..Or similar Line

We have corrected this sentence

“Siva et al demonstrated the feasibility of preserving lung functional volumes in a cohort of 14 patients treated with 3D conformal radiotherapy for NSCLC adapted to 68Ga-perfusion PET/CT”

253: “The group from the Peter MacCallum Cancer Centre in Melbourne, Australia, was the first and only so far that assessed lung PET/CT imaging for functional lung avoidance” • it would be better to add the reference of SIVA [19] at this point? And repeat it at line 256? If readers don't know that Siva is from Peter MacCallum they seems two different experiences with only one reference.

We have clarified this sentence

“Siva et al. were the first and only so far that assessed lung PET/CT imaging for functional lung avoidance. In a series of 14 patients with lung cancer irradiated in conformal technique, they demonstrated that PET-guided functional planning allowed a reduction by a median of 0.86 Gy of the MLD in well perfused lungs”

line 85: write the definition of OAR because it's the first time you cite OAR in the paper

We have written the definition of OAR

“SBRT is a more precise and conformal technique that allows the irradiation of smaller volumes of organs at risk (OARs) »

line 110 specify “total activity” (total counts? Or other measures?)

“Total counts” was changed for “total integrated counts »

Line 133: define ITV because it's the first time you cite it

We have defined ITV

Line 151: delete were and write “was carried out”.

We have corrected this mistake

Line 153: mean lung dose (MLD) please choose only one extended definition or acronym because you have already used them

We do not understand this remark because we have used only one acronym and one definition

line 158: please define which are the “median differences” and specify their calculation

We have defined the “median differences” and specified their calculation

“Median absolute (dosimetric parameter of functional plan minus dosimetric parameter of anatomical plan) and relative (dosimetric parameter of functional plan minus dosimetric parameter of anatomical plan divided by dosimetric parameter of anatomical plan) differences and their respective range were reported.”

Table 1 Please re-aline on the left the first column

We have re-aligned the first column

Table 2 structure is misleding making difficult to understand results

We have changed the structure

Line 244-246: please can you simplify to make text clear

We have simplified the text

“By incorporating an additional "functional lung volume constraint" to the lung FV50% in addition to the standard constraints in SBRT planning”

Line 249: “As expected, the absolute gains obtained with SBRT, ranging from 0.17 to 0.22 Gy, were lower than with conformal techniques” • May be a reference necessary?

We have added a reference

As expected, the absolute gains obtained with SBRT, ranging from 0.17 to 0.22 Gy, were lower than with conformal techniques due to lower healthy lung volume irradiated in SBRT [2].

line 265: is the cited reference correct?

We have corrected this mistake,  it is the number 14

Line 282: delete were, write are

We have corrected this mistake

line 297: delete min, write minutes

We have corrected this mistake

line 297-300: “Furthermore, with one 68Ga-MAA synthesis, 6 68Ga Perfusion PET/CT scans can be performed. These make the test very convenient to implement in a daily PET/CT program, especially during the hour between the first injections of FDG and the first acquisition on the camera”. • It's not so clear for all readers how is managed the workflow of PET facilities • may be better “ Furthermore, one synthesis production of 68Ga-MAA allows to perform perfusion PET/CT for 6 patients, making the test very convenient to implement in a daily PET/CT program. This advantage is especially evident during the hour (uptake time) between the first injections of 18FFDG, the most used tracer in oncology, and its first acquisition on the PET tomograph.

We have changed the sentence

Furthermore, one synthesis production of 68Ga-MAA allows to perform 68Ga-perfusion PET/CT for 6 patients, making the test very convenient to implement in a daily PET/CT program. This advantage is especially evident during the hour (uptake time) between the first injections of 18F-FDG, the most used tracer in oncology, and its first acquisition on the PET tomograph.

line 302: “due to growing use for prostate cancer and neuroendocrine tumour imaging” • may be better “ due to growing use of 68Ga-based tracers for prostate cancer and neuroendocrine tumour imaging”

We have changed the sentence

Finally, 68Ga-MAA is very convenient radiotracer for clinical use and 68Ga generators are increasingly available in nuclear medicine departments due to growing use of 68Ga-based tracers for prostate cancer and neuroendocrine tumour imaging.

line 303-321: there are some repetitions. Can you simplify?

We have simplified this paragraph

“Our study has several limitations. Firstly, we only assessed the feasibility of personalizing radiation planning to lung functional mapping using 68Ga-perfusion PET/CT. However, it would be value to evaluate the prognostic impact in terms of RILI of doses delivered to the lung functional volumes. Indeed, a meta-analysis including 6 studies showed that the dose to functional volume had better predictive values than the dose to anatomical volume, including V20, V5 and MLD [14]. But, all these studies used SPECT and none of the studies was statistically significant. Secondly, by adding “lung functional volume” constraints, the functional planning led to a significant reduction of the dose to the functional volumes but also to the lung anatomical volume. Nevertheless, there was a greater improvement in the dose to the functional volumes as compared with that of the anatomical lung, showing that the optimization is not blinded but functionally adapted. Of note is that the reduction of the dose in the AV was not reported in most of studies assessing imaging techniques for lung functional avoidance [14]. Thirdly, we aimed to deliver the same doses to the OARs for the functional planning as for the anatomical planning. However, doses to OAR were significantly lower than the constraints in the most cases, leaving the possibility of improved optimization to further preserve lung functional volumes.”

Line 329-332: a little convoluted wording. Can you make clearer? Beacause these are conclusions

We have changed this sentence to make more clearer

“This is the first study to evaluate and demonstrate the benefits of 68Ga perfusion PET/CT in the lung SBRT planning process in preserving functional lung volumes while respecting target volume coverage and doses to other OARs.”

Figure 1. Example of anatomic volume (AV) and lung functional volumes (FV) delineation. The FV50%, FV70% and FV90% volumes were defined as the minimal volume containing 50% (FV50%), 70% (FV70%) and 90% (FV90%) of the total activity within the AV. The first column of coronal images is not described and starting from the left, the the first volume is 90%. Please can you better describe images?

We have detailed the description of figure 1.

Figure 1. Example of anatomic volume (AV) (at the left) and lung functional volumes (FV) delineation. The FV90% (in blue), FV70% (in green) and FV50% (in red) volumes were defined as the minimal volume containing 90% (FV90%), 70% (FV70%) and 50% (FV50%) of the total activity within the AV.

Appendix A: PET/CT protocol. Line 352 Move the sentence “The patients were instructed to breathe freely for the duration of the scans. “ after the first one “The patients were positioned supine on the PET/CT scanner with their arms raised. “

We have reversed these 2 sentences

line 353-358: CT and PET protocol may be better described It would be interesting if you have any data about quality controls of the prepared tracer to add in the Appendix A.

As proposed,  data about quality controls were added in appendix A.

[68Ga]Ga-MacroAggregatted Albumin (MAA) suspensions were prepared in the radiopharmacy using an automated process with a miniAIO® module and disposable cassettes from Trasis (Belgium). The fully automated process was performed from a commercial MAA kit used for 99mTc labelling suspended in sodium acetate solution and a 68Ga eluate without pre purification as previously described by Blanc-Béguin et al.  The obtained suspensions were GMP compliant. Controls performed on [68Ga]Ga-MAA suspensions from independent syntheses were : radio-chemical purity > 98.6%, particles < 3µm : <2.2%, radionuclidic purity > 99,999%, pH = 4, endotoxins level < 5 IU/mL and all suspensions were sterile.

Reviewer 2 Report

My main concern is your approach to data analyses.  Since the demographics are provided in nonparametric style and your figures suggest the data are not normally distributed, I do not understand why you used parametric statistical comparisons.  In Table 2, the descriptor "Mean dose" should be removed.  I understand that the data in this table are presented as medians and range.

Since each patient likely served as their own control, why was not a paired nonparametric statistical test approach used?

Why was not a nonparametric statistical analysis used for all of your data?

Line 179 - I don't believe that 99.56 is different than 99.55.  Since you have only 60 patients, all percentage data presentation is valid only to integer accuracy.

Please consult a statistician to assist you with optimal analysis of the data arising from this clinical investigation.  I suspect reanalysis will result in a revised presentation of your Results and Discussion.

Author Response

"My main concern is your approach to data analyses.  Since the demographics are provided in nonparametric style and your figures suggest the data are not normally distributed, I do not understand why you used parametric statistical comparisons. 

In Table 2, the descriptor "Mean dose" should be removed.  I understand that the data in this table are presented as medians and range.

We removed "mean" in table 2

Since each patient likely served as their own control, why was not a paired nonparametric statistical test approach used?
Why was not a nonparametric statistical analysis used for all of your data?
Line 179 - I don't believe that 99.56 is different than 99.55.  Since you have only 60 patients, all percentage data presentation is valid only to integer accuracy.
Please consult a statistician to assist you with optimal analysis of the data arising from this clinical investigation.  I suspect reanalysis will result in a revised presentation of your Results and Discussion."

Thank you for your comment on the statistical test to use.

We have reviewed our results with statisticians.

Initially, we had used a parametric test for paired data because some studies suggest that it is possible to use this type of test in the case of a non-normal distribution with a population greater than 30.

However, the distribution was significantly non-normal, so we used the non-parametric Wilcoxon test for paired data as you suggested.

The only modification concerning our results concerns the Dmax at the heart which is not statistically different with this test.

The dose at PTV is still significant (p=0.01). We exchanged with statisticians to understand this result while the medians are very close (99.6 vs 99.6). A possible explanation is the higher statistical power obtained by using paired data tests with slightly lower values for a majority of patients in the functional planning group.

Moreover, we have changed our results by reporting only one significant number.

We have removed the following sentences from the results and discussion

« There was also a statistically significant difference regarding the maximum dose to the heart 5.65 Gy (0.17-38.94 Gy) vs 6.62 Gy (0.16-40.00 Gy) (p=0.03) (dose constraint <30 Gy, 32 Gy and 40 Gy for 3, 4 and 8 fractions, respectively). »

« Finally, the improvements obtained in the lung functional volume resulted in a statistically significant increase (5.65 vs 6.62 p=0.03) in the maximum dose to the heart »

« However, the doses delivered to the heart were well below the dose constraints »